# Quantifying the fitness effects of resistance alleles with and without anthelmintic selection pressure using *Caenorhabditis elegans*

**Amanda O. Shaver**[1,2], **Isabella R. Miller**[1], **Etta S. Schaye**[1], **Nicolas D. Moya**[2],
**J. B. Collins**[1,2], **Janneke Wit**[1], **Alyssa H. Blanco**[1], **Fiona M. Shao**[1], **Elliot J. Andersen**[1],
**Sharik A. Khan**[1], **Gracie Paredes**[1], **Erik C. Andersen**[2]*

**1** Molecular Biosciences, Northwestern University, Evanston, Illinois, United States of America, **2** Dept. of Biology, Johns Hopkins University, Baltimore, Maryland, United States of America

* erik.andersen@gmail.com

**Data Availability Statement:** All code and data used to replicate the data analysis and figures are

## Abstract

Albendazole (a benzimidazole) and ivermectin (a macrocyclic lactone) are the two most commonly co-administered anthelmintic drugs in mass-drug administration programs worldwide. Despite emerging resistance, we do not fully understand the mechanisms of resistance to these drugs nor the consequences of delivering them in combination. Albendazole resistance has primarily been attributed to variation in the drug target, a beta-tubulin gene. Ivermectin targets glutamate-gated chloride channels (GluCls), but it is unknown whether GluCl genes are involved in ivermectin resistance in nature. Using *Caenorhabditis elegans*, we defined the fitness costs associated with loss of the drug target genes singly or in combinations of the genes that encode GluCl subunits. We quantified the loss-of-function effects on three traits: (i) multi-generational competitive fitness, (ii) fecundity, and (iii) development. In competitive fitness and development assays, we found that a deletion of the beta-tubulin gene *ben-1* conferred albendazole resistance, but ivermectin resistance required the loss of two GluCl genes (*avr-14* and *avr-15*). The fecundity assays revealed that loss of *ben-1* did not provide any fitness benefit in albendazole conditions and that no GluCl deletion mutants were resistant to ivermectin. Next, we searched for evidence of multi-drug resistance across the three traits. Loss of *ben-1* did not confer resistance to ivermectin, nor did loss of any single GluCl subunit or combination confer resistance to albendazole. Finally, we assessed the development of 124 *C. elegans* wild strains across six benzimidazoles and seven macrocyclic lactones to identify evidence of multi-drug resistance between the two drug classes and found a strong phenotypic correlation within a drug class but not across drug classes. Because each gene affects various aspects of nematode physiology, these results suggest that it is necessary to assess multiple fitness traits to evaluate how each gene contributes to anthelmintic resistance.

available at https://github.com/AndersenLab/bzml_manuscript.

**Funding:** This work was supported by the National Institutes of Health NIAID grant R01AI153088 and a National Science Foundation Capacity grant 2224885 granted to ECA. The funders had no role in study design, data collection and analysis, decision to publish, or preparation of the manuscript.

**Competing interests:** The authors have declared that no competing interests exist.

## Author summary

Control of parasitic nematodes often depends on mass-drug administration (MDA) programs, where combinations of anthelmintics are distributed to at-risk populations. Two commonly co-administered anthelmintic drugs in MDA programs are albendazole and ivermectin, and resistance to both drugs has emerged. Although the mechanism of resistance (MoR) to albendazole has been primarily attributed to variation in a beta-tubulin gene, the MoR to ivermectin remains unknown. Ivermectin acts through pentameric glutamate-gated chloride channels (GluCls). However, it is unclear whether genes that encode GluCls are involved in ivermectin resistance in parasitic nematodes. Using *Caenorhabditis elegans*, we quantified the fitness costs associated with deletions of the beta-tubulin gene *ben-1* and three genes encoding GluCl subunits *avr-14*, *avr-15*, and *glc-1* on three traits: (i) multi-generational competitive fitness, (ii) fecundity, and (iii) development. We found different anthelmintic responses across strains and traits but no evidence of multi-drug resistance. Our results suggest that multiple traits should be considered to understand resistance comprehensively and that the determination of whether a gene plays a role in anthelmintic resistance depends on the trait measured. Understanding the quantitative effects and fitness-associated costs of each GluCl subunit in *C. elegans* can help explain the costs of mutations in these subunits in parasites.

## Introduction

Parasitic nematodes are some of the most abundant, diverse, and destructive parasites of humans that cause significant socio-economic and health impacts, including the collective loss of over eight million disability-adjusted life years (DALYs) [1–3]. Control of these parasites often depends on mass-drug administration (MDA) programs, where anthelmintics are distributed in combinations to at-risk populations. However, anthelmintic resistance has emerged in veterinary helminths, with reports of reduced drug efficacy against nematodes in humans, which threatens our ability to control parasitic nematode infections [4–8]. In veterinary medicine, overuse of anthelmintics has placed strong selective pressures on parasites, which has led to the evolution of resistance to all major drug classes [9–11] and highlights the potential for a similar pattern of anthelmintic resistance to spread throughout human parasitic nematode populations.

Simultaneous treatment with two or more drugs (*i.e.*, co-administration) from different anthelmintic classes is one method proposed to slow the development of resistance [12–15]. Anthelmintic rotation, another suggested method to slow resistance, uses the periodic switching of drug classes to alleviate selection pressures on one drug class and prolong drug lifespan and efficacy [16–18]. Despite the implementation of both strategies, co-administration and anthelmintic rotation have failed to control resistance and pose the risk of multi-drug resistance [9,19,20], particularly if a shared mechanism confers resistance to both drugs (*e.g.*, drug export) [21], a phenomenon known as cross-resistance [22]. Although empirical evidence for cross-resistance is lacking, its potential occurrence could increase the development of resistance to both drugs. To date, various accounts of multi-drug resistance have been reported in veterinary medicine [21–24]. Therefore, it is critical to define and address the mechanisms of resistance (MoR) for each drug in a treatment plan to ensure that drug efficacy can be reached and to prevent multi-drug resistance.

Two of the most commonly used anthelmintic drugs in MDA programs are albendazole, a benzimidazole (BZ), and ivermectin, a macrocyclic lactone (ML) [23–26], where research

suggests that these two drugs can be co-administered as a chemotherapeutic treatment for intestinal helminths and lymphatic filariasis [27–30]. Because albendazole and ivermectin are two drugs on the World Health Organization (WHO) Model Lists of Essential Medicines [27,30–32], it is critical to identify the MoR for both drugs to inform appropriate administration and slow the development of resistance. However, anthelmintic resistance in parasites can be difficult to disentangle because of multiple factors, including a lack of access to relevant life stages and *in vitro* culture systems, dependence on vertebrate hosts, and a limited molecular toolkit (*e.g.*, gene knockouts and induced mutations cannot be used to study genes associated with resistance in parasites) [33]. With its ease of growth, outstanding genetic tractability, and molecular toolkits, the free-living nematode *Caenorhabditis elegans* has contributed to the identification and characterization of the Mode of Action (MoA) and MoR of all major anthelmintic drug classes [33–40]. Additionally, wild *C. elegans* strains from the *Caenorhabditis* Natural Diversity Resource (CaeNDR) [41] have been used to explore anthelmintic resistance in natural populations and to uncover novel MoR [36,41–48].

Both the laboratory-adapted strain, N2, and *C. elegans* wild strains have facilitated the identification and characterization of the beta-tubulin gene *ben-1* as the primary target of albendazole and other BZs [34,35,45,46,49]. Loss-of-function mutations in *ben-1* have been identified in *C. elegans* strains resistant to BZs [50]. Furthermore, resistance alleles corresponding to point mutations in *ben-1* homologs in parasitic nematode populations continue to be identified [35,49,51,52]. Notably, the redundancy among the six beta-tubulin genes in *C. elegans* allows strains with a non-functional *ben-1* gene to develop normally [50]. To date, beta-tubulins have been the most well characterized anthelmintic target across nematodes [50,53–58].

Ivermectin acts as a positive allosteric modulator that selectively opens inhibitory glutamate-gated chloride channels (GluCls) in the membranes of pharyngeal muscles, motor neurons, female reproductive tracts, and the excretory/secretory pores [59–62]. However, the relationship between GluCls and the MoR of ivermectin is poorly understood. To date, mutations in GluCl subunits have been infrequently detected and associated with ivermectin-resistant parasitic nematodes. Our ability to find GluCl mutations in nature can be complicated by the variation in GluCl gene family number across species, interactions between genes outside the GluCl family, poor nematode genome quality leading to false negatives, and the locations of GluCl tissue expression (see *Discussion*).

In *C. elegans*, GluCls are thought to be homopentameric or heteropentameric transmembrane complexes where six genes encode GluCl subunits: *avr-14*, *avr-15*, *glc-1*, *glc-2*, *glc-3*, and *glc-4* [63–66]. Although it is established that GluCl subunits are the main targets of ivermectin in *C. elegans*, null mutations in *avr-14*, *avr-15*, or *glc-1* individually do not cause ivermectin resistance [66]. However, mutagenesis studies of *C. elegans* showed that a triple GluCl mutant strain (*avr-14; avr-15 glc-1*) displayed greater than 4000-fold resistance and that a double GluCl mutant strain (*avr-14; avr-15*) had intermediate levels of resistance as compared to the wild-type strain [66]. This study demonstrated that mutations in multiple GluCl subunit genes can cause high-level ivermectin resistance. Nevertheless, it is essential to note that the mutagenesis studies in *C. elegans* were not performed in a controlled background and assessed only one trait, survival, as measured by placing embryos on agar plates with ivermectin and observing the number of embryos that grew to adulthood [64,66]. Because many genes, as shown by others [66–74] and outlined below, can cause ivermectin resistance, a controlled genetic background is essential. The Dent study performed screens in 10 ng/ml of ivermectin on Nematode Growth Medium (NGM) plates with animals in an *avr-15* mutant background.

By comparing the Dent study with our current investigation, it is essential to note several key methodological differences. The Dent study performed $EC_{37}$ estimates, which encompassed a range of ivermectin concentrations from 0.6 ng/ml (0.5 nM) to 13.8 ng/ml (12 nM)

for strains without Dyf mutations. Our plate-based assays were conducted at 1.5 nM ivermectin, which does fall within the range of ivermectin concentrations used in the Dent assays. However, the Dent study allowed embryos to grow to adulthood for two weeks, which would allow developmentally delayed animals to reach maturity as *C. elegans* completes its life cycle in four days. Moreover, assessing multiple traits to evaluate ivermectin resistance is important because GluCls are widely expressed across several tissue types in the *C. elegans* nervous system and pharynx [67,75]. Survival is an aggregate trait that integrates developmental rate, fecundity, numerous behaviors, and metabolism. To adequately evaluate the MoR to ivermectin, fitness, development, and fecundity should be used to understand how GluCl subunit genes interact and play a role in resistance and, ultimately, assess how they could affect the spread of resistance alleles in parasite populations.

Using *C. elegans*, we defined the fitness costs associated with the loss of *ben-1*, *avr-14*, *avr-15*, and *glc-1* and the loss of combinations of GluCl subunits all in a controlled genetic background on nematode resistance to albendazole and ivermectin. We focused on the three genes encoding GluCl subunits that have been most implicated in ivermectin resistance to date: *avr-14*, *avr-15*, and *glc-1* [64,66,67,75–77]. We measured three fitness components: (i) multi-generational competitive fitness, (ii) fecundity, and (iii) development. First, in the competitive fitness assay, we found that loss of *ben-1* conferred albendazole resistance, and loss of GluCl subunits did not confer albendazole resistance. We found that loss of *avr-15* carried significant fitness consequences when not under drug selection pressure. Under constant ivermectin exposure, loss of both *avr-14* and *avr-15* and three GluCl subunits (*avr-14*, *avr-15*, and *glc-1*) caused strong selective advantages compared to the wild-type strain. Second, in the fecundity assays, we found that loss of *ben-1* did not confer any advantage in the presence of albendazole and that all strains with an *avr-15* deletion had reduced fecundity in all conditions compared to the wild-type strain. Third, in our assessment of development, we found that loss of *ben-1* conferred resistance to albendazole, and loss of both *avr-14* and *avr-15* or three GluCl subunits (*avr-14*, *avr-15*, and *glc-1*) conferred ivermectin resistance. Fourth, we sought to identify any evidence of cross-resistance between albendazole and ivermectin by comparing the fitness costs of each deletion mutation at *ben-1*, *avr-14*, *avr-15*, and *glc-1* in both drugs. Across the three fitness traits we assessed, we found that the *ben-1* deletion mutant strain did not confer resistance in the presence of ivermectin, nor did the GluCl deletion mutant strains display resistance in the presence of albendazole. Fifth, we assessed the development of 124 *C. elegans* wild strains across six BZs and seven MLs to identify evidence of cross-resistance between the two drug classes in natural populations. We found a strong correlation with phenotype within a drug class but not across drug classes, which indicates that phenotypic responses to the two drug classes are distinct, likely because they target different aspects of nematode development. Here, we present a comprehensive study that assessed the quantitative effects that *ben-1* and GluCl mutations have on various aspects of nematode fitness in the presence of albendazole or ivermectin. These results suggest that conclusions about a gene's involvement in anthelmintic resistance depend on the trait assessed and that multiple fitness traits must be considered to understand resistance comprehensively.

## Results

### Multi-generational competitive fitness assays show how loss of beta-tubulin and GluCl subunits are selected in control or anthelmintic conditions

CRISPR-Cas9 genome editing was performed to generate four deletion strains that each cause loss of function. Each strain contains a single deletion in either the beta-tubulin gene, *ben-1*, or the GluCl subunit genes *avr-14*, *avr-15*, or *glc-1* (**Fig 1**, **S1 Table**). Next, to comprehensively

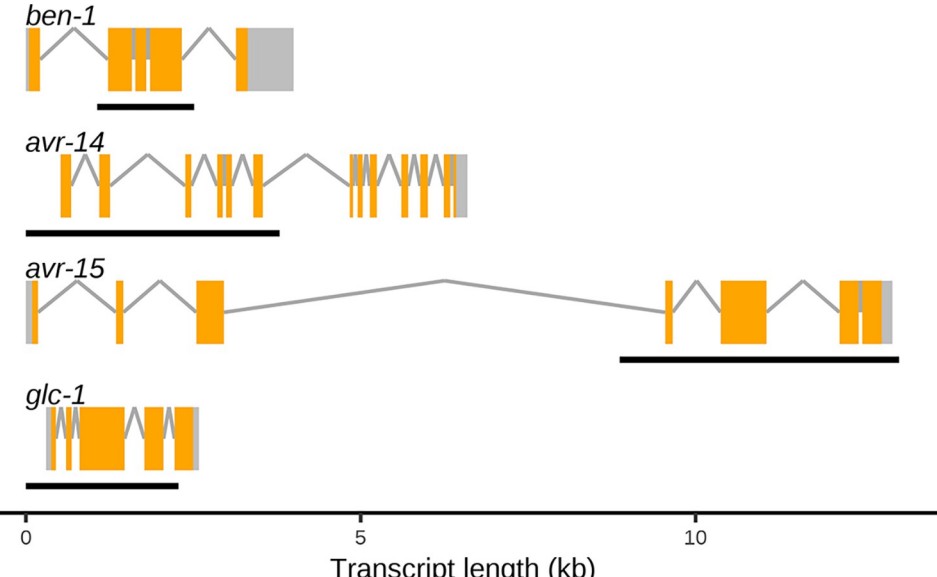

**Fig 1. Gene models of *ben-1* and the three genes encoding for GluCl subunits in *C. elegans*.** Predicted gene models presented for *ben-1*, *avr-14*, *avr-15*, and *glc-1* include exons (orange rectangles) and introns (gray lines) in the *C. elegans* laboratory-adapted strain N2 background (WS283). Black bars underneath each gene display the span of the deletion present in each gene for the strains assayed (**S1 Table**).

assess the role that *avr-14*, *avr-15*, and *glc-1* play in ivermectin resistance, we created double mutants of each combination of GluCl deletion alleles and a triple mutant (*avr-14; avr-15 glc-1*) by crossing the single deletion strains (see *Methods*). We performed competitive fitness assays to determine the selective advantages or disadvantages of alleles in control or drug treatment conditions. Fitness involves the ability of an organism, or population, to survive and reproduce in its environment [78,79]. In these assays, a query strain was competed against the barcoded wild-type strain PTM229 (**S1 Table**), which contains a synonymous change in the *dpy-10* locus in the N2 background and does not cause any fitness effects compared to the normal laboratory N2 strain [80] in the presence of dimethyl sulfoxide (DMSO), albendazole, or ivermectin (**S1 Fig**). Because all query strains contain the wild-type *dpy-10* locus, allele frequencies of *dpy-10* between PTM229 and each query strain were measured for each generation to quantify relative fitness.

The competitive fitness assays enabled us to focus on two key traits critical to nematode fitness: time to reproduction and reproductive rate. These assays allow us to observe small effects on nematode fitness over multiple generations. If an allele confers a deleterious fitness effect compared to the wild-type allele, then that strain will decrease in frequency over the generations. Conversely, if an allele confers a beneficial effect compared to the wild-type allele, then that strain will increase in frequency over the generations. Finally, if an allele has no difference in effect when compared to the wild-type allele, then the two strains will be found at approximately equal frequencies throughout the competitive fitness assay.

In control conditions, the wild-type strain, N2, showed no differences in competitive fitness compared to the barcoded wild-type control strain, PTM229, as expected. The loss of *ben-1*, *avr-14*, or *glc-1* in single deletion mutants, along with the loss of both *avr-14* and *glc-1* in the double mutant strain, did not have significant differences in competitive fitness as compared to the control strain, which suggests that a deletion in these genes, in control conditions, did not cause fitness consequences (**Figs 2A, 2B and S2**). Notably, all strains with a loss of *avr-15*,

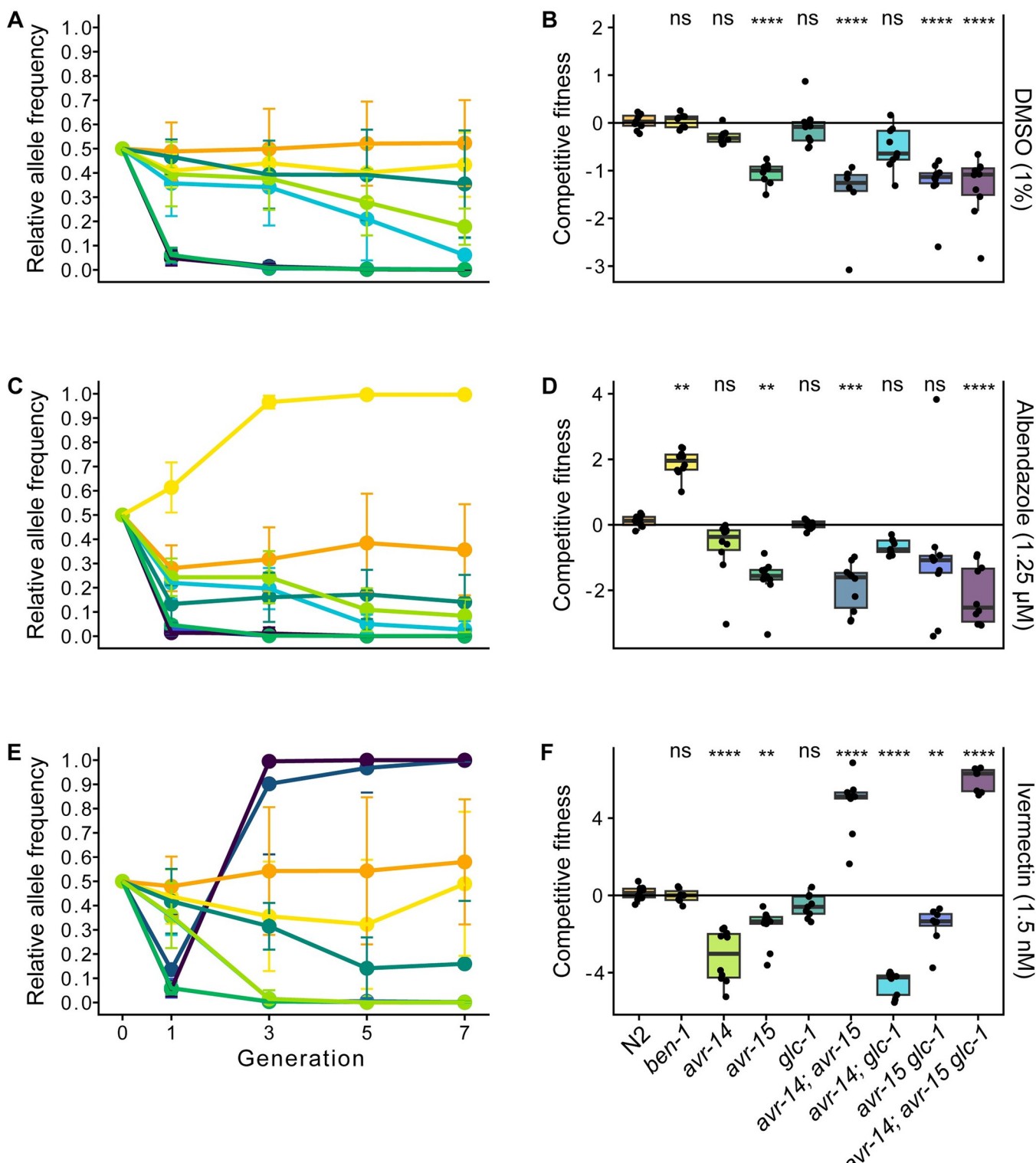

**Fig 2. Competitive fitness assays across seven generations in DMSO, albendazole, and ivermectin.** A barcoded N2 wild-type strain, PTM229, was competed with strains that have deletions in either one, two, or three genes that encode GluCl channels or the beta-tubulin gene *ben-1* in **(A)** DMSO (1%), **(C)** albendazole (1.25 μM), and **(E)** ivermectin (1.5 nM). Generation is shown on the x-axis, and the relative allele frequencies of the nine strains with genome-edited alleles and N2 are shown on the y-axis. The $\log_2$-transformed competitive fitness of each allele is plotted in **(B)** DMSO (1%), **(D)** albendazole (1.25 μM), and **(F)** ivermectin (1.5 nM). The gene tested is shown on the x-axis, and the competitive fitness is shown on the y-axis. Each point represents a biological replicate of that competition experiment. Data are shown as Tukey box plots with the median as a solid horizontal line, and the top and bottom of the

box representing the 75th and 25th quartiles, respectively. The top whisker is extended to the maximum point that is within the 1.5 interquartile range from the 75th quartile. The bottom whisker is extended to the minimum point that is within the 1.5 interquartile range from the 25th quartile. Significant differences between the wild-type N2 strain and all the other alleles are shown as asterisks above the data from each strain ($p > 0.05$ = ns, $p < 0.001$ = \*\*\*, $p < 0.0001$ = \*\*\*\*, Tukey HSD). Because two DMSO competitive fitness assays were performed, results from both DMSO assays are reported (**S2 Fig**).

whether it be a single, double, or triple GluCl mutant, were vastly unfit and ceased to exist in the population by the third generation (**Fig 2A**). These results suggest that, when not under drug selection pressure, loss of *avr-15* is severely detrimental to animal fitness. Hence, we hypothesize that GluCl loss-of-function alleles are not anticipated to occur within parasite populations.

In drug conditions, we can assess whether loss of any of these genes causes resistance. In albendazole conditions, the loss of *ben-1* caused a strong fitness advantage over the control strain and swept to fixation in the population by the third generation (**Fig 2C and 2D**). Notably, the loss of one, two, or all three GluCl genes caused a reduction of fitness in albendazole conditions, and each strain displayed a similar competitive fitness response in albendazole as observed in control conditions (**Fig 2C and 2D**). In ivermectin conditions, the loss of *ben-1* or *glc-1* caused no significant differences in competitive fitness as compared to the control strain. The loss of *avr-14* or *avr-15*, the loss of both *avr-14* and *glc-1*, or the loss of both *avr-15* and *glc-1* caused significantly reduced competitive fitness in ivermectin compared to the control strain (**Fig 2E and 2F**). By contrast, the loss of both *avr-14* and *avr-15* and the loss of all three GluCl genes (*avr-14*, *avr-15*, and *glc-1*) caused strains to sweep to fixation in the presence of ivermectin by the third generation, indicating that they had significantly improved fitness as compared to the control strain (**Fig 2E**). Results from these competitive fitness assays suggest that ivermectin resistance is only observed when both *avr-14* and *avr-15* functions are lost. The fitness disadvantages of losing *avr-14* or *avr-15* alone outweigh any ivermectin resistance that could be present. The loss of *glc-1* in addition to loss of both *avr-14* and *avr-15* caused the triple mutant strain to sweep to fixation faster than the *avr-14; avr-15* double mutant strain. However, these two strains did not significantly differ in competitive fitness.

## Fecundity effects caused by loss of beta-tubulin or GluCl subunits in control or anthelmintic conditions

To dissect the genetic basis of anthelmintic resistance, we must identify the roles of *ben-1* and the GluCl subunit genes in nematode fecundity and their potential influence on the spread and persistence of resistance alleles in a population. As a measure of relative fitness, fecundity refers to the number of offspring produced by an organism [78,79]. To compare the effects on fecundity caused by the loss of *ben-1* and the GluCl subunit genes, we measured lifetime fecundity and daily fecundity of the nine *C. elegans* strains (see *Methods*). Single L4 larval stage hermaphrodites were placed on nematode growth medium (NGMA) plates under control (DMSO), albendazole, or ivermectin conditions [81]. Hermaphrodites were transferred every 24 hours for five days and maintained under standard laboratory conditions. After five days, hermaphrodites were transferred for the final time to a new NGMA plate for 48 hours. We manually counted the offspring from images of assay plates from single hermaphrodites. The results showed considerable differences in lifetime fecundity among the nine strains across the three conditions (**Fig 3**).

Fecundity directly impacts population growth rate and survival. Measuring fecundity provides insights into reproductive success, which is fundamental to the assessment of animal fitness and resilience when exposed to environmental changes. Here, the fecundity assays enabled us to focus on offspring production both in control conditions and under constant drug selection pressure. By assessing fecundity in control (DMSO) conditions, we can discern

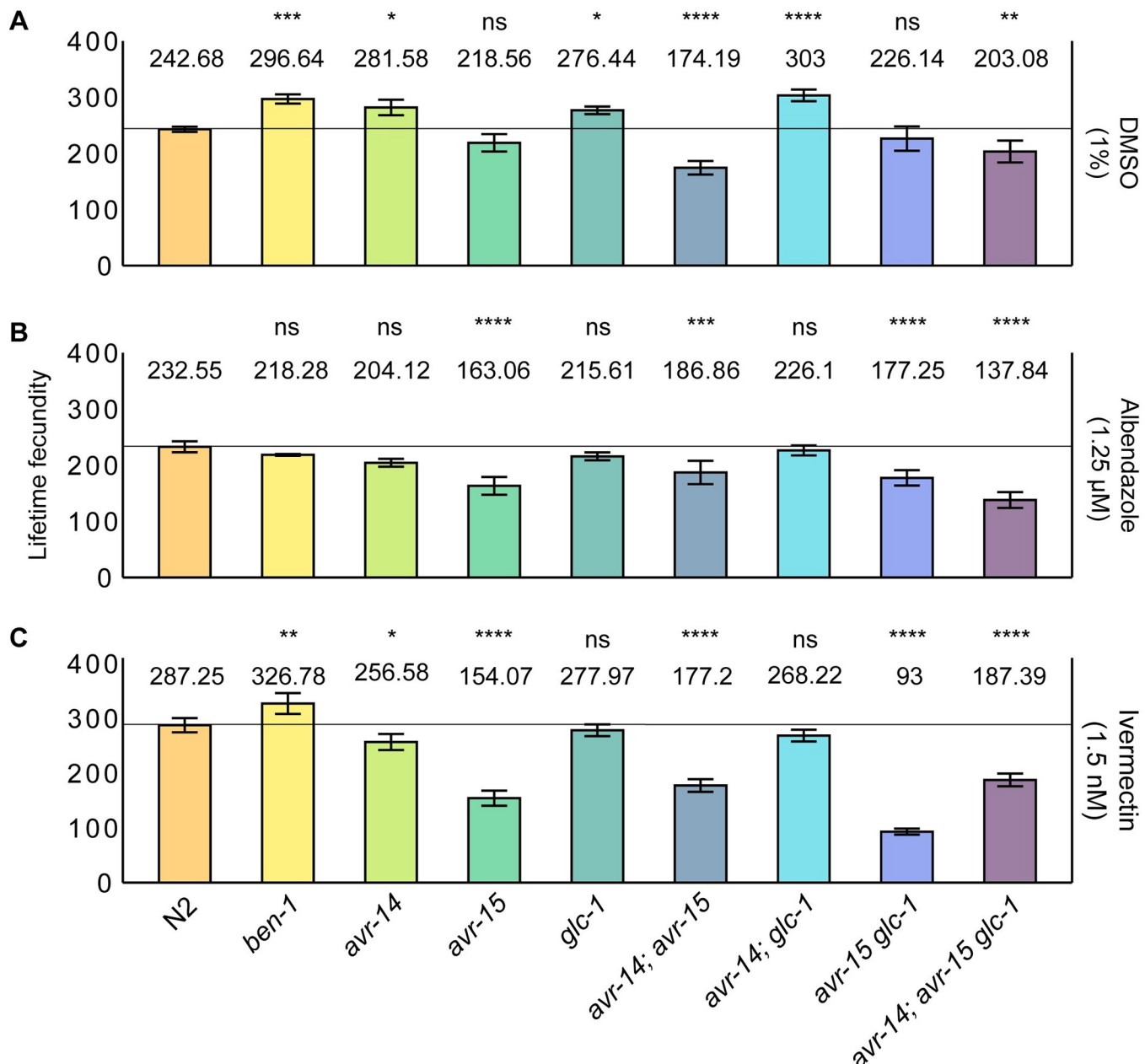

**Fig 3. Variation in lifetime fecundity of beta-tubulin and GluCl mutants in the presence of DMSO, albendazole, or ivermectin.** Bar plots for lifetime fecundity, y-axis, for each deletion strain on the x-axis in **(A)** DMSO (1%), **(B)** albendazole (1.25 μM), and **(C)** ivermectin (1.5 nM) are shown. Error bars show the standard deviation of lifetime fecundity among 7–10 replicates. The laboratory reference strain, N2, is colored orange. Other strains are colored by genotype. Comparisons of lifetime fecundity between the laboratory reference strain, N2, and all deletion strains are shown. Statistical significance was calculated using Tukey HSD. Significance of each comparison is shown above each comparison pair ($p > 0.05$ = ns, $p < 0.05$ = *, $p < 0.01$ = **, $p < 0.001$ = ***, $p < 0.0001$ = ****, Tukey HSD).

the fitness effects caused by loss of *ben-1* or by the loss of GluCl subunit genes when not under drug selection pressure. The loss of *ben-1* caused increased fecundity (**Fig 3A**), which suggests that *ben-1* limits fecundity in control conditions. The loss of *avr-15* alone and loss of both *avr-15* and *glc-1* did not affect lifetime fecundity as compared to the N2 strain. However, the loss of *avr-15* in combination with *avr-14* or in combination with *avr-14* and *glc-1* caused

significantly reduced lifetime fecundities as compared to the N2 strain, which suggests that in combination *avr-14* and *avr-15* or *avr-14*, *avr-15*, and *glc-1* are necessary for normal fecundity in control conditions (**Fig 3**). By contrast, the loss of *avr-14* or *glc-1* and the loss of both genes caused significantly increased lifetime fecundities as compared to the N2 strain, which suggests that *avr-14* and *glc-1* limit fecundity in control conditions.

By evaluating fecundity under drug conditions, we can uncover the fitness effects caused by mutations in *ben-1* or the GluCl subunit genes and determine how these mutations under drug pressure could affect the spread of potential resistance alleles in a population. In the presence of albendazole, the loss of *ben-1* did not cause a significant effect on lifetime or daily fecundity as compared to the N2 strain, which suggests that *ben-1* does not confer albendazole sensitivity by alteration of nematode fecundity (**Figs 3B and S4**). However, in the presence of ivermectin, the loss of *ben-1* had a significant increase in lifetime fecundity compared to the control strain, exhibiting the same pattern observed in control conditions (**Figs 3C and S5**). The loss of *glc-1* and the loss of both *avr-14* and *glc-1* did not cause significant differences in lifetime or daily fecundity compared to the N2 strain in albendazole or ivermectin, which suggests that *glc-1* alone or in combination with *avr-14* are necessary for normal fecundity production under drug pressure. By contrast, a loss of *avr-14* alone caused significantly reduced fecundity compared to the control strain in ivermectin, which indicates that *avr-14* is necessary for fecundity in ivermectin. Additionally, the loss of *avr-15* alone, the loss of both *avr-14* and *avr-15* or *avr-15* and *glc-1*, or the loss of all three GluCl subunits caused significantly reduced lifetime fecundity compared to the control strain, a trend observed across all conditions, which suggests that *avr-14* and *avr-15* alone or in combination are necessary for normal fecundity. Although the loss of *avr-15* in the single, double, or triple mutant strains caused a reduction in fecundity across conditions, the daily fecundity patterns varied across the three conditions (**S3–S5 Figs**). In all three conditions, strains with a loss of *avr-15* had a reduction in daily fecundity between days two and three compared to the N2 strain. In DMSO and ivermectin, strains with a loss of *avr-15* had an increase in daily fecundity between days five and seven at the end of the assay (**S4 Fig**). Because strains with a loss of both *avr-14* and *avr-15* and a loss of all three GluCl subunits have significantly reduced fecundity across all conditions, it would be unlikely for animals in nature to acquire loss-of-function mutations in these genes that cause detrimental fitness consequences.

## Loss of *ben-1* conferred albendazole resistance, and loss of both *avr-14* and *avr-15* or all three GluCl subunits conferred ivermectin resistance

We then performed high-throughput assays (HTAs) to measure nematode length, a proxy for development, in strains with a loss of *ben-1* or loss of GluCl subunit genes (**S2 Table**) in response to drug treatment. The assay included 72 replicates per strain with 5–30 animals per replicate in each drug or control condition. The reported nematode length of each strain is the delta between animal lengths in control and drug conditions to obtain normalized animal length and assess drug effects. Longer median animal length (*i.e.*, larger animals) than the N2 strain corresponds to increased resistance to the tested drugs, and shorter median animal length (*i.e.*, smaller animals) than the N2 strain corresponds to increased sensitivity to the tested drugs. Strains varied in length after growth for 48 hours in control conditions, but the loss of *avr-14* and *avr-15* caused the most significant delays in development (**S6 Fig**). Despite substantial variation among strains in control (DMSO) conditions, animal measurements were categorized as the L4 larval stage by our custom CellProfiler worm models [82,83], indicating all strains underwent normal development in control conditions.

As previously reported, the N2 strain was developmentally delayed in albendazole, where animals were shorter than in control conditions, demonstrating sensitivity to albendazole [66].

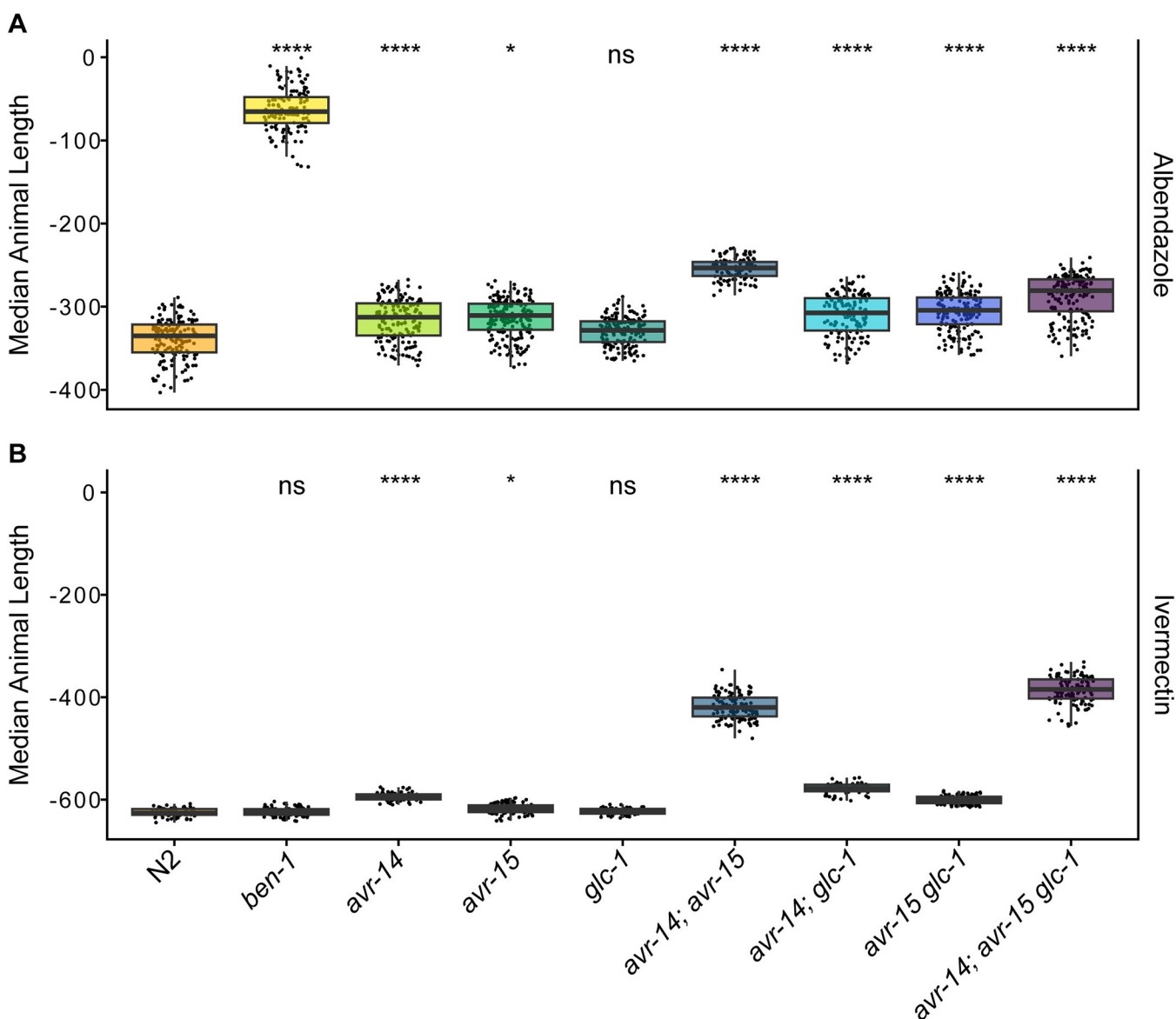

**Fig 4. High-throughput assays for each mutant strain in the presence of albendazole and ivermectin.** The regressed median animal length values for populations of nematodes grown in either **(A)** 30 μM albendazole or **(B)** 250 nM ivermectin are shown on the y-axis. Each point represents the normalized median animal length value of a well containing approximately 5–30 animals. Data are shown as Tukey box plots with the median as a solid horizontal line, and the top and bottom of the box representing the 75th and 25th quartiles, respectively. The top whisker is extended to the maximum point that is within the 1.5 interquartile range from the 75th quartile. The bottom whisker is extended to the minimum point that is within the 1.5 interquartile range from the 25th quartile. Significant differences between the wild-type strain and all other deletions are shown as asterisks above the data from each strain ($p > 0.05$ = ns, $p < 0.001$ = ***, $p < 0.0001$ = ****, Tukey HSD).

By contrast, the loss of *ben-1* caused albendazole resistance, as demonstrated by longer animal length than observed in the N2 strain (**Fig 4A**). Although each GluCl deletion mutant strain had significantly longer animal lengths than the N2 strain, none of the seven GluCl deletion mutant strains conferred albendazole resistance as observed in the *ben-1* deletion mutant (**Fig 4A**). In ivermectin, the N2 strain had the greatest delay in development. Although, a loss of *avr-14* or *avr-15* alone or a loss of *avr-14* and *glc-1* or *avr-15* and *glc-1* in combination had significantly longer median animal lengths than the N2 strain in ivermectin (**Fig 4B**).

However, the loss of *avr-14* and *avr-15* or the loss of *avr-14*, *avr-15*, and *glc-1* caused quantitative ivermectin resistance as compared to the N2 strain. This ivermectin resistance confirms previous findings [66]. However, we did not see a significant difference in median animal length between the double GluCl mutant *avr-14* and *avr-15* or the triple GluCl mutant, as reported previously [49,75]. A higher concentration of ivermectin (500 nM) was also measured and confirmed the same patterns described above (**S7 Fig**).

## No evidence of cross-resistance or multi-drug resistance between BZs and MLs

Because albendazole and ivermectin can be distributed together to at-risk populations to treat lymphatic filariasis and has been proposed as a strategy to treat soil-transmitted helminth infections [28–30,84,85], it is critical to ensure that the two drugs do not have the same MoR to avoid the possibility of cross-resistance. Because we compared the response of the *ben-1* deletion mutant strain in ivermectin and all GluCl deletion mutant strains in albendazole for all fitness assays, we obtained a comprehensive picture of how these genes interact in the presence of a drug that is not designed to affect their given target. In the competitive fitness and HTAs, the loss of *ben-1* did not cause ivermectin resistance. Additionally, none of the GluCl deletion mutant strains conferred resistance to albendazole across the competitive fitness and HTAs, as compared to a loss of *ben-1*. However, it is important to note that a loss of *ben-1* did confer a slight advantage compared to the N2 strain in the ivermectin fecundity assay (**Fig 3C**). All GluCl deletion mutant strains, except the *glc-1* deletion mutant strain, conferred a slight advantage compared to the N2 strain in the albendazole HTA (**Fig 4A**). However, it is essential to note that these slight advantages across conditions could be caused by genes targeted by both drugs expressed in the same tissues or cells. Importantly, the competitive fitness assays did not display any evidence of cross-resistance (**Fig 2**).

Our fitness assays showed how *ben-1*, *avr-14*, *avr-15*, and *glc-1* respond under drug pressure, but we know that these genes do not account for all of the albendazole or ivermectin resistance found across the *C. elegans* species [49,86]. Therefore, we performed an HTA (see *Methods*) to assess the nematode development of 124 wild strains in the presence of six BZs and seven MLs, which included albendazole and ivermectin. We used a Spearman's Rank correlation test to identify any evidence of multi-drug resistance among the BZs and MLs. We find much stronger phenotypic correlations of responses within the same drug class than we do between drug classes (**Fig 5**), and no significant correlations exist across the two drug classes (**S3 Table**), which suggests that we did not detect evidence of multi-drug resistance between the two drug classes.

## Discussion

Here, we assessed multiple fitness traits to understand how *ben-1* and three genes that encode GluCl subunits contribute to albendazole and ivermectin resistance. Additionally, the quantitative assessment of *avr-14*, *avr-15*, and *glc-1* in ivermectin response is critical to understand how GluCl subunits affect fitness in nematode populations. Because *C. elegans* shares the major characteristics of the parasitic nematode body plan, such as the cuticle and organization of the nervous system, along with a conserved neuromuscular system and neurotransmitters [33,87], the traits that we assessed can help us better understand how resistance alleles could spread in parasitic nematode populations.

Our competitive fitness and development assay results confirm previous findings, which showed that a loss of *ben-1* confers albendazole resistance and that a loss of both *avr-14* and *avr-15* are necessary to confer ivermectin resistance [75,77,78]. Animals with the loss of both

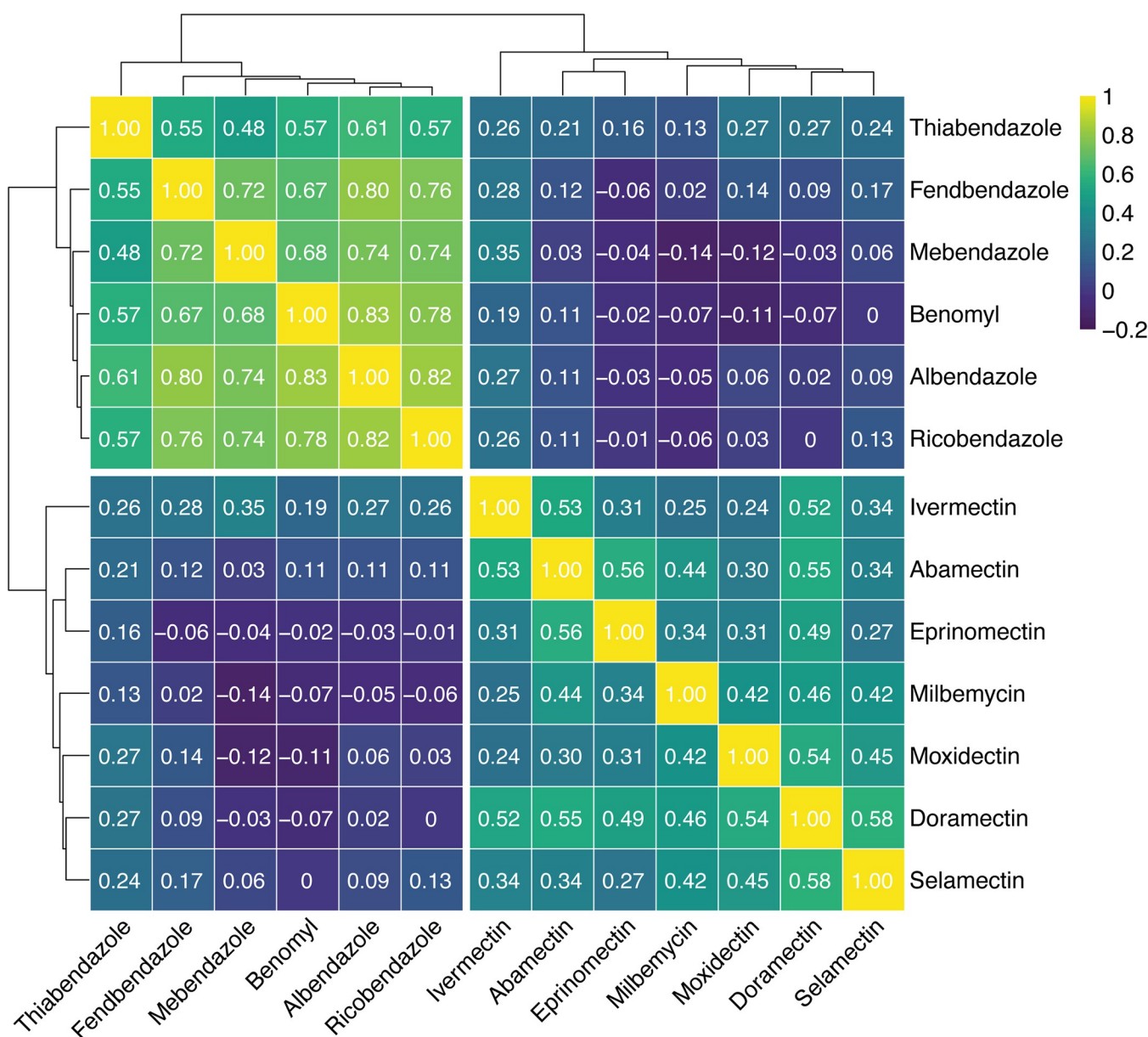

**Fig 5.** Spearman-rank correlations between 124 wild isolates exposed to BZs and MLs. Spearman-rank correlations and significance testing were performed between 124 wild isolates exposed to six BZs and seven MLs. The dendrograms were constructed using Euclidean distance and complete linkage metrics and then split into their two largest clusters to show the relationships of similarities between the 124 wild strains exposed to the two major anthelmintic classes. A correlation of 1 signifies the strongest phenotypic correlation (*i.e.*, identical median animal lengths) and a correlation of -0.2 signifies the weakest phenotypic correlation (*i.e.*, dissimilar median animal lengths). Significant correlations ($p < 0.05$) were recorded (S3 Table).

*avr-14* and *avr-15* or the loss of three GluCl subunits (*avr-14*, *avr-15*, and *glc-1*) did not significantly differ in ivermectin response as previously reported in mutagenesis studies [82,84,85]. It is also critical to note that strains with a loss of both *avr-14* and *avr-15* or a loss of three GluCl subunits (*avr-14*, *avr-15*, and *glc-1*) have significant fitness consequences when not under ivermectin exposure, rendering it unlikely that loss-of-function mutations will occur in these genes and confer ivermectin resistance in nature.

On the contrary, animals under strong selection because of prolonged ivermectin exposure might select resistance alleles in GluCl genes, which would allow mutations in GluCl subunits to spread throughout populations. However, parasitic nematodes must remain under constant ivermectin exposure at higher than therapeutic doses to overcome the severe fitness consequences that mutations in GluCl subunits carry for selection to remain significant. Because human drug treatment programs apply ivermectin once to three times annually [88], it is unlikely that ivermectin persists in humans between treatments at concentrations that would cause constant drug selection pressure and subsequent selection of resistance alleles in parasitic nematodes. Although veterinary medicine prescribes ivermectin more frequently, ivermectin again is unlikely to remain at concentrations high enough to cause selective pressures, especially during winter months when treatment subsides [89,90].

Prior to this study, minimal research had been performed to assess the quantitative contributions of GluCl subunits on different traits involved in nematode fitness. To identify why GluCls in parasites lack mutations in ivermectin-resistant populations [91] and why we have yet to validate ivermectin-resistant genes, we tested the contributions of three genes that encode GluCl subunits on *C. elegans* fitness. We found that the competitive fitness and development assays produced similar results, but the fecundity assays deviated from the results of these two assays. We have two hypotheses to explain this discrepancy between assays. First, the competitive fitness assay focuses on two traits involved in nematode fitness: development (*i.e.*, time to reproduction) and reproductive rate. Because competitive fitness encompasses development, we can expect results from the competitive fitness assay and development assay to be more comparable than the results of the fecundity assay. In the fecundity assay, we measured the daily and total fecundity of individual animals over seven days and did not focus on reproductive rate. Therefore, it is difficult to compare the daily fecundity of animals for the duration of one generation to the competitive fitness assay, where animals have offspring over multiple generations. Second, it is possible that mutations in *ben-1* or the GluCl subunit genes do not affect fecundity to the same extent that they affect development, rendering the fecundity results less clear to interpret compared to the competitive fitness and development assays.

We found that a loss of *avr-14* alone or in combination with *glc-1* has significant fitness consequences in ivermectin but not when in albendazole or control conditions. Overall, a loss of *avr-15* has profoundly detrimental effects compared to the control strain across all conditions. An animal with a loss of *avr-15* in combination with *avr-14* must remain under constant ivermectin pressure to exhibit any fitness benefit compared to the control strain. Finally, a loss of *glc-1* does not affect fitness across conditions, indicating perhaps why mutations of this gene are found in *C. elegans* wild populations [86,92,93]. Our investigation did not reveal any indication that *glc-1* is responsible for conferring resistance to ivermectin, which contradicts earlier findings [92,93]. However, we focused on the effects of muti-generational competitive fitness, fecundity, and development when nematodes are exposed to ivermectin. By contrast, previous studies measured nematode response against different macrocyclic lactones (*i.e.*, abamectin), focused on different traits such as body bends, paralysis, and gene expression, and used different genetic approaches to assess resistance [92,93]. The discrepancy between our findings and earlier research underscores the necessity of evaluating multiple traits, as the response to anthelmintic treatment can vary depending on the trait measured.

Several reasons might explain the infrequent detection of GluCl subunits mutated in parasitic nematode isolates that have ivermectin resistance, including variation in GluCl gene family number, interactions between genes outside the GluCl family, poor nematode genome quality, and the locations of GluCl tissue expression. First, we do not fully understand the composition of GluCls. The GluCls are members of the pentameric ligand-gated ion channel family and, similar to other members of this family, the functional channels formed *in vivo* could

be homomeric or heteromeric [89]. More research needs to be done to elucidate the subunit composition of GluCls across parasitic nematode species. In addition, no auxiliary proteins, analogous to the genes that influence the trafficking and assembly of nematode nicotinic acetylcholine receptors, have been reported for the GluCls, and we do not know if any such proteins exist [91]. Furthermore, although *avr-14*, *avr-15*, *glc-1*, *glc-2*, *glc-3*, and *glc-4* are predicted to be conserved widely across nematode species [94], *avr-14* is currently the only GluCl gene found to be highly conserved throughout *Nematoda* [26,92–94]. A polymorphism in an *avr-14* ortholog has been identified in *Cooperia oncophora* [57]. Although polymorphisms in several genes (*Hco-glc-3* and *Hco-glc-5*) have been identified in resistant isolates of *Haemonchus contortus* [26,95–97], recent genome-wide analyses have not identified mutations in GluCl subunit associated in ivermectin resistant in *H. contortus* [98]. Importantly, these results are correlative and do not show a causal connection between ivermectin resistance and GluCl genes. Additionally, research in both *C. elegans* and parasitic nematodes has led to the suggestion that ivermectin resistance might be polygenic [94], so combinations of genes must also be considered.

Second, the absence of mutations in GluCl genes in ivermectin-resistant parasites has led researchers to search for additional genes outside this gene family involved in the MoR of ivermectin [91]. It is unknown why, despite GluCls being understood as the MoA and confirmation of association with resistance to ivermectin in *C. elegans*, the GluCl subunits have not been widely associated with resistance in parasite populations. As illustrated here, the fitness disadvantages of losing *avr-14* or *avr-15* alone or in combination outweigh any ivermectin resistance. Additionally, the loss of *glc-1* alone or in combination with *avr-14* or *avr-15* had no discernible impact on fitness across conditions but also did not confer ivermectin resistance. Hence, unlike alleles that confer albendazole resistance, alleles associated with ivermectin resistance exhibit pronounced fitness consequences. Consequently, susceptible alleles would be more likely to endure within parasite populations because they confer fitness advantages in the absence of ivermectin. Our findings provide insights into why GluCls have not been clearly associated with resistance in parasites.

To date, ivermectin resistance across *Nematoda* appears to involve various genes and mechanisms [66,68–70]. Additional genes outside of those that encode for GluCl subunits are implicated in the MoR of ivermectin, which includes Dyf genes and genes involved in neuronal development and function, such as *unc-7*, *unc-9*, *unc-38*, and *unc-63* [98–101]. Genes involved in ivermectin metabolism, such as ATP-binding cassette (ABC) transporters, cytochrome P450 enzymes, GABA receptors, P-glycoproteins, and other signaling proteins, have been implicated in ivermectin resistance though much more research needs to be done to determine their role in the MoR of ivermectin [67,72–74]. Ivermectin also affects some nicotinic receptors and acts as a positive allosteric modulator of the α7 neuronal nicotinic acetylcholine receptor [99]. Overall, additional genes outside of the GluCl family could be involved in the MoR of ivermectin. However, functional studies, such as those conducted in the present study, are essential to elucidate the role each gene plays in ivermectin resistance.

Third, it is important to highlight that GluCl subunits have been well characterized in only a few nematode species, partly because of poor-quality genomes. Recent efforts have been made to generate high-quality reference parasitic nematode genomes [100–103]. WormBase Parasite [104,105] serves as the main repository for these data, which now hosts a collection of 240 genomes, representing 181 species. Recently, 864 total GluCl gene predictions were categorized across 125 species into orthologous groups, which suggests that additional GluCl subunits across *Nematoda* have yet to be discovered [94]. As our genome assemblies, technologies, and analytical techniques improve, so will our ability to search for and identify GluCl genes.

Fourth, although albendazole and ivermectin target different genes and have different MoR, it is conceivable that the genes targeted by both drugs can be expressed in the same tissues or cells. To date, our understanding of the MoR and tissue-specific susceptibility for most anthelmintic drugs across nematode species is not well known. Previous research has shown that *ben-1* function in neurons, specifically in cholinergic neurons, underlies susceptibility to BZs [105]. However, analogous experiments are imperative to ascertain in which tissues GluCl subunit genes underlie ivermectin susceptibility. Expression data from the Complete Gene Expression Map of the *C. elegans* Nervous System (CeNGEN) [106] shows an overlap in *ben-1* and GluCl subunit gene expression in neurons (**S8 Fig**). In particular, a pronounced overlap between *ben-1* and *avr-14* expression in cholinergic neurons is observed, which suggests albendazole and ivermectin could target the same tissues. The overlap in *ben-1* and GluCl gene expression in the same neuronal cell subtypes could explain the small advantages that *ben-1* conferred in ivermectin and the GluCls in albendazole in the HTA (**Fig 4**). The co-expression of two genes in the same neurons implies a potential functional relationship between the genes, which could collaborate to regulate specific neural functions associated with the neurotransmitter. A shared expression between genes could lead to unexpected consequences, such as changes in sensitivity to other drugs or alterations in neural processes beyond resistance to the targeted drug. Given that multiple GluCls are present in *C. elegans*, functional redundancy is conceivable. Because neurotransmitters play essential roles in physiological processes, including behavior, locomotion, and sensory perception, it is imperative to delineate which neurons are implicated in GluCl expression and consequently influences ivermectin susceptibility. Fully understanding the MoR of each drug class is complicated by the implication that genes associated with drug resistance overlap in expression within the same neuronal pathways. Finally, it is important to note that all the expression data discussed here have been performed in *C. elegans* and not parasitic nematode models, so differences among nematodes might not be captured entirely by research on this free-living nematode species.

In summary, our experiments suggest that loss of *ben-1* confers albendazole resistance and that multiple mutations in GluCl genes are required to obtain ivermectin resistance. Nevertheless, our understanding of ivermectin's MoR remains incomplete because we have not identified all the genes involved in ivermectin resistance in *C. elegans* nor identified the genes involved in ivermectin resistance in parasitic nematodes. To solve this problem, we need to conduct additional experiments that quantitatively assess the fitness effects of all six GluCl subunit genes singly and in combination, both in control and ivermectin conditions. Moreover, to identify in which tissues GluCl function underlies ivermectin susceptibility, transgenic strains that express each GluCl subunit genes in different tissues will determine the tissue-specific susceptibility of ivermectin. Finally, when considering the opportunity for multi-drug resistance to occur, either by cross-resistance or independent selection, our data suggest that cross-resistance is unlikely because *ben-1* and the GluCl subunits do not appear to target the same MoR in *C. elegans*. In studies of *H. contortus*, multi-drug resistant isolates carry independent mutations in genes specific to a given resistance phenotype, suggesting that albendazole and ivermectin target different genes [98]. Studies across *C. elegans* and *H. contortus* suggest that animals acquire multi-drug resistance independently and not by cross-resistance.

## Materials and methods

### Generation of *ben-1* and GluCl deletion strains

Nine *Caenorhabditis elegans* deletion strains generated from an N2 background were used in this study (**S1 Table**). The *ben-1* deletion mutant strain was generated in the N2 background using CRISPR-Cas9 genome editing [49,107]. The *avr-14*, *avr-15*, and *glc-1* single deletion

mutant strains were generated in the PD1074 background using CRISPR-Cas9 genome editing by SunyBiotech (Fujian, China). The PD1074 strain is a lineal descendent of the laboratory-adapted N2 Bristol strain and is >99.97% genetically identical to the N2 strain, with 116 structural discrepancies between the two strains [108]. Over 98% of the N2 genes encode unchanged products in PD1074 [108]. The N2 and PD1074 strains are grouped into the same isotype because of the high degree of genetic similarity [41,109]. The double and triple GluCl deletion mutant strains were generated by crossing the single GluCl deletion strains.

## *C. elegans* strains and maintenance

In the competitive fitness assays, the barcoded wild-type strain PTM229 *dpy-10* (*kah81*) was used as a control strain. The PTM229 strain is an N2 strain that contains a synonymous change in the *dpy-10* locus that does not have any growth effects compared to the normal laboratory N2 strain [104]. Animals were maintained at 20˚C on 6 cm plates with modified nematode growth medium (NGMA), which contains 1% agar and 0.7% agarose to prevent animals from burrowing. The NGMA plates were seeded with the *Escherichia coli* strain OP50 as a nematode food source. All strains were grown for three generations without starvation on NGMA plates before anthelmintic exposure to reduce the transgenerational effects of starvation stress. The specific growth conditions for nematodes used in each assay are described below.

## Nematode food preparation for NGMA assays

A batch of OP50 *E. coli* was grown and used as a nematode food source for each competitive fitness and fecundity assay. A frozen stock of OP50 *E. coli* was streaked onto a 10 cm Luria-Bertani (LB) agar plate and incubated overnight at 37˚C. The following day, a single bacterial colony was transferred into two culture tubes that contained 5 ml of 1x LB. The starter cultures and two negative controls (1X LB without *E. coli*) were incubated for 18 hours at 37˚C shaking at 210 rpm. The $OD_{600}$ value of the starter cultures were measured using a spectrophotometer (BioRad, SmartSpec Plus) to calculate how much starter culture was needed to inoculate a one-liter culture at an $OD_{600}$ value of 0.005. For each assay, one culture contained one liter of pre-warmed 1X LB inoculated with the starter culture that grew for approximately 4–4.5 hours at 37˚C at 210 rpm. Cultures were grown until they reached an $OD_{600}$ value between 0.45 and 0.6. Cultures were transferred to 4˚C to suspend growth. OP50 was spotted on NGMA test plates (two per culture) and grown at 37˚C overnight to ensure a normal lawn was grown with no contamination.

## Nematode food preparation for liquid culture assays

One batch of HB101 *E. coli* was used as a nematode food source for all HTAs. A frozen stock of HB101 *E. coli* was streaked onto a 10 cm LB agar plate and incubated overnight at 37˚C. The following day, a single bacterial colony was transferred into three culture tubes that contained 5 ml of 1x Horvitz Super Broth (HSB). The starter cultures and two negative controls (1X HSB without *E. coli*) were incubated for 18 hours at 37˚C shaking at 180 rpm. The $OD_{600}$ value of the starter cultures were measured using a spectrophotometer (BioRad, SmartSpec Plus) to calculate how much starter culture was needed to inoculate a one-liter culture at an $OD_{600}$ value of 0.001. A total of 14 cultures each of which contained one liter of pre-warmed 1X HSB inoculated with the starter culture grew for 15 hours at 37˚C while shaking at 180 rpm until cultures were in the late log growth phase. After 15 hours, flasks were removed from the incubator and transferred to 4˚C to arrest growth. The 1X HSB was removed from the cultures through three rounds of washing and centrifugation, where the supernatant was removed, and the bacterial cells were pelleted. Bacterial cells were washed, resuspended in K medium, pooled, and

transferred to a 2 L glass beaker. The $OD_{600}$ value of the bacterial suspension was measured and diluted to a final concentration of $OD_{600}$100 with K medium, aliquoted to 15 ml conical tubes, and stored at -80˚C for use in the HTAs.

## Anthelmintic dose-response assays

Before selecting anthelmintic doses to use in the competitive fitness and fecundity assays, a dose response was performed to determine the appropriate dosage of albendazole and ivermectin for the given assay. For the dose-response assays, gravid hermaphrodites were picked and placed into a bleach spot. After 24 hours, L1 larvae were picked to a new 6 cm NGMA plate. After 48 hours, ten late-stage L4 larvae hermaphrodites were picked to NGMA plates in the below conditions. Animals were exposed to albendazole at the following concentrations (μM): 0 (1% DMSO), 0.65, 1.25, and 2.5. Animals were exposed to ivermectin at the following concentrations (nM): 0 (1% DMSO), 1.2, 1.3, 1.4, 1.5, 1.6, 1.7, 1.8, 1.9, 2, 2.1, 2.25, 2.5, 5, and 10. Animals were imaged every day for seven days. After seven days in each condition, NGMA plates were checked for starvation. The highest concentration of each condition that allowed the most sensitive strain (**S9 Fig**) to starve after seven days (*e.g.*, 1.5 nM ivermectin) compared to the most resistant strain (**S10 Fig**) was selected to use in competitive fitness and fecundity assays. Using the described strategy to select the anthelmintic dosage allowed sensitive strains to compete against the wild-type control strain, PTM229, in the competitive fitness assays. The same anthelmintic dosage was used in the competitive fitness and fecundity assays to keep all NGMA-based assays consistent.

## Anthelmintic stock preparation for competitive fitness assays, fecundity assays, and HTAs

Albendazole and ivermectin stock solutions were prepared with DMSO (Fisher Scientific, Catalog # D1281). Albendazole (Sigma-Aldrich, Catalog # A4673-10G) was used at a concentration of 1.25 μM in the competitive fitness and brood size assays and 30 μM in the HTA. Ivermectin (Sigma-Aldrich, Catalog # I8898-1G) was used at a concentration of 1.5 nM in the competitive fitness and brood size assays and 250 nM and 500 nM in the HTA. Anthelmintic stock solutions were prepared, aliquoted, and stored at -20˚C for use in the assays (**S4 Table**). All control conditions for every assay exposed animals to 1% DMSO, a concentration shown not to affect *C. elegans* normal physiology and lifespan [110,111].

## Competitive fitness assays

We used previously established pairwise competitive fitness assays to assess nematode fitness [107]. The fitness of a strain was determined by comparing the allele frequency of a test strain against the allele frequency of the wild-type control strain PTM229. Strains contain molecular barcodes to distinguish between the two strains using oligonucleotide probes complementary to each barcoded allele. Ten L4 larval individuals of each strain were placed onto a single 6 cm NGMA plate along with ten L4 larval individuals of the PTM229 strain. Ten independent NGMA plates of each competition were prepared for each strain in each condition: control (DMSO), albendazole (1.25 μM), or ivermectin (1.5 nM). The N2 strain was included to ensure that assays were reproducible and that all plates had effective albendazole and ivermectin concentrations. Plates were grown for roughly one week to starvation (*i.e.*, one generation). Animals were transferred to a new NGMA plate of the same condition by the transfer of a 0.5 cm$^3$ NGMA piece from the starved plate onto the new plate. The remaining individuals on the starved plate were washed into a 15 mL Falcon tube with M9 buffer, concentrated by centrifugation, transferred to 1.5 mL Eppendorf tubes, and stored at -80˚C. Competitions were

performed for seven generations, and animals were collected after generations one, three, five, and seven. DNA was extracted in randomized blocks using the DNeasy Blood & Tissue kit (Qiagen, Catalog # 69506), purified with the Zymo DNA cleanup kit (Catalog # D4064), and diluted to approximately 1 ng/μL.

We quantified the relative allele frequency of each strain as previously described [81,108,109]. A droplet digital PCR (ddPCR) approach with TaqMan probes (Applied Biosciences) was used. Using TaqMan probes, the ddPCR assay was performed with a Bio-Rad QX200 device with standard probe absolute quantification settings. The TaqMan probes selectively bind to wild-type *dpy-10* and the *dpy-10* allele present in PTM229 [110]. Thresholds were manually selected and set in QX Manager software (Version 2.1). Relative allele frequencies of each tested allele were calculated using the QuantaSoft software. Calculations of relative fitness were calculated by linear regression analysis to fit the data to a one-locus generic selection model [111] (**S5** and **S6 Tables**).

## Fecundity assays

Brood size assays were used to assess nematode fecundity for the laboratory-adapted strain, N2, and the eight mutant strains. Prior to each assay, strains were grown for three generations at 20°C to reduce cross-generational effects. For each *C. elegans* strain in the fourth generation, single L4 larval stage hermaphrodites were picked to each of ten 6 cm NGMA plates spotted with OP50 and were maintained at 20°C. Ten independent 6 cm NGMA plates seeded with *E. coli* OP50 were prepared for each strain in each condition, control (DMSO), albendazole (1.25 μM), and ivermectin (1.5 nM) and maintained at 20°C. For each assay plate, the original hermaphrodite parent was transferred to a fresh plate every 24 hours for 96 hours. A custom-built imaging platform (DMK 23GP031 camera; Imaging Source, Charlotte, NC, USA) was used to collect images for each of the first four assay plates (0, 24, 48, and 72-hour assay plates) 48 hours after the removal of the parent from each NGMA plate. Images of the fifth assay plates were collected 72 hours after the final transfer of the parents. The total offspring were counted from each image by visual inspection using the use of the Multi-point tool in ImageJ (Version 1.54f) [112]. The original hermaphrodite parents were excluded from the counts. The number of offspring in each of the first four assay plates corresponds to daily fecundity (**S3, S4 and S5 Figs**). The number of offspring on the fifth assay plates contained offspring from three days (days 5–7). For each biological replicate of each *C. elegans* strain, the lifetime fecundity was calculated as the total number of offspring from the five plates (**Fig 3**). Replicates where the original hermaphrodite parent died were excluded from the analysis of lifetime fecundity [81,113,114]. Only biological replicates with data from all five assay plates were used to calculate daily and total fecundity (**S7 Table**).

## High-throughput assays (HTAs) to assess nematode development

Populations of each strain were amplified and bleach-synchronized for three independent assays. Independent bleach synchronizations controlled for variation in embryo survival and subsequent effects on developmental rates that could be attributed to bleach effects. After bleach synchronization, approximately 30 embryos were dispensed into the wells of a 96-well microplate in 50 μL of K medium. Three 96-well plates were prepared per bleach for each strain. Each 96-well microplate was prepared, labeled, and sealed using gas-permeable sealing films (Fisher Scientific, Catalog # 14-222-043). Plates were placed in humidity chambers to incubate overnight at 20°C while shaking at 170 rpm (INFORS HT Multitron shaker). After 24 hours, every plate was inspected to ensure that all embryos hatched and animals were developmentally arrested at the first larval (L1) stage so all strains started each assay at the same

developmental stage. Next, food was prepared to feed the developmentally arrested L1 animals using the required number of $OD_{600}100$ HB101 aliquots (see *Nematode food preparation for liquid culture assays*). The aliquots were thawed at room temperature, combined into a single conical tube, and diluted to an $OD_{600}30$ with K medium. To inhibit further bacterial growth and prevent contamination, 150 μM of kanamycin was added to the HB101. Working with a single drug at a time, an aliquot of anthelmintic stock solution was thawed at room temperature (see *Anthelmintic stock preparation*) and diluted to a working concentration. The anthelmintic working concentration was set to the concentration that would give the highest desired dose when added to the 96-well microplates at 1% of the total well volume. The dilution of the anthelmintic working solution was prepared using the same diluent, DMSO, used to make the stock solution. The anthelmintic dilution was then added to an aliquot of the $OD_{600}30$ K medium at a 3% volume/volume ratio. Next, 25 μl of the food and anthelmintic mixture was transferred into the appropriate wells of the 96-well microplates to feed the arrested L1s at a final HB101 concentration of $OD_{600}10$ and expose L1 larvae to the given anthelmintic. Immediately afterward, the 96-well microplates were sealed using a new gas permeable sealing film, returned to the humidity chambers, and incubated for 48 hours at 20˚C shaking at 170 rpm. The remaining 96-well microplates were fed and exposed to anthelmintics in the same manner. After 48 hours of incubation and shaking in the presence of food and anthelmintic, the 96-well microplates were removed from the incubator and treated with 50 mM sodium azide in M9 for 10 minutes to paralyze and straighten nematodes. After 10 minutes, images of nematodes in the microplates were immediately captured using a Molecular Devices ImageXpress Nano microscope (Molecular Devices, San Jose, CA) using a 2X objective. The ImageXpress Nano microscope acquires brightfield images using a 4.7 megapixel CMOS camera and stores images in a 16-bit TIFF format. The images were used to quantify the development of nematodes in the presence of anthelmintics as described below (see *High-throughput imager assay [HTA] data collection* and *data cleaning*).

## High-throughput assay (HTA) data collection and data cleaning

The CellProfiler software program (Version 4.0.3) was used to characterize and quantify biological data from the image-based assays. Custom software packages designed to extract animal measurements from images collected on the Molecular Devices ImageXpress Nano microscope were previously described [83]. CellProfiler modules and Worm Toolbox were developed to extract morphological features of individual *C. elegans* animals from images from the HTA [115]. Worm model estimates and custom CellProfiler pipelines were written using the WormToolbox in the GUI-based instance of CellProfiler [116]. Next, a Nextflow pipeline (Version 20.01.0) was written to run command-line instances of CellProfiler in parallel on the Quest High-Performance Computing Cluster (Northwestern University). The CellProfiler workflow can be found at (https://github.com/AndersenLab/cellprofiler-nf). The custom CellProfiler pipeline generates animal measurements by using four worm models: three worm models tailored to capture animals at the L4 larval stage, in the L2 and L3 larval stages, and the L1 larval stage, respectively, as well as a "multi-drug high dose" (MDHD) model, to capture animals with more abnormal body sizes caused by extreme anthelmintic responses. These measurements composed our raw dataset.

Data analysis steps have been modified from previous reports [43,117]. All analyses were performed using the R statistical environment (version 4.2.1) unless stated otherwise. The HTA produced hundreds of images per experimental block; thus, we implemented a systematic approach to assess the quality of animal measurement data in each well. Several steps were implemented to clean the raw image data using metrics indicative of high-quality animal measurements for downstream analysis.

1) Objects with a *Worm_Length* > 100 μm were removed from the CellProfiler data to (A) retain L1 and MDHD-sized animals and (B) remove unwanted particles [118]. By using the *Worm_Length* > 100 μm threshold to retain small sensitive animals, more small objects, such as debris, were also retained [116].

2) R/*easyXpress* [83] was used to filter measurements from worm objects within individual wells with statistical outliers and to parse measurements from multiple worm models down to single measurements for single animals.

3) The data were visualized by anthelmintic, anthelmintic concentration, assay, strain, and worm model for two purposes. First, to ensure that each strain, by assay, contained control wells that had a *mean_wormlength_um* between 600–800 μm, the size of an L4 animal. If the *mean_wormlength_um* in the control wells was not between the 600–800 μm range, then that strain and/or assay were removed. This filter ensured the control, DMSO, wells primarily contained L4 animals. Second, we wanted to identify whether albendazole or ivermectin contained a high abundance of MDHD model objects across all assays and concentrations. Anthelmintics with an abundance of objects classified by the MDHD model across assays and concentration likely contain debris. We then reduced the data to wells that contained between five and thirty animals, under the null hypothesis that the number of animals is an approximation of the expected number of embryos originally titered into wells (approximately 30). Given that our analysis relied on well median animal length measurements, we excluded wells with less than five animals to reduce sampling error.

5) Next, we removed measurements from each anthelmintic drug that were no longer represented in at least 80% of the independent assays because of previous data filtering steps or had fewer than five measurements per strain.

Importantly, all four steps described above were implemented to ensure all data were cleaned (*i.e.*, outliers removed), and we obtained balanced assays and complete strain and condition representation for all strains in the HTA. Using this process, we retained data from every assay, bleach, and condition for each strain, which highlights that all strains have appropriate representation in the analysis (**S2 Table**).

5) Finally, we normalized the data by (1) regressing variation attributable to assay and technical replicate effects and (2) normalizing these extracted residual values to the average control phenotype. For each anthelmintic drug, we estimated a linear model using the raw phenotype measurement as the response variable and both assay and technical replicate identity as explanatory variables following the formula *median_wormlength_um ~ assay + bleach* (which accounts for bleach effects during bleach synchronization) using the *lm()* function in base R. We then extracted the residuals from the linear model for each anthelmintic and subtracted normalized phenotype measurements in each anthelmintic from the mean normalized phenotype in control (DMSO) conditions. These normalized phenotype measurements were used in all downstream statistical analyses.

## Wild Strain HTA and Spearman rank-order correlations

Populations of 124 *C. elegans* wild strains were processed using the HTA as described above (see *High-throughput assays [HTAs]*). Each wild strain was exposed to the following benzimidazoles or macrocyclic lactones at the denoted concentrations: abamectin (2 nM) (Millipore sigma, Catalog # 31732), albendazole (10.65 μM) (Fluka, Catalog # A4673-10G), benomyl (20.66 μM) (Sigma Aldrich, Catalog # 45339-250MG), doramectin (5 nM) (Millipore Sigma,

Catalog # 33993), eprinomectin (44 nM) (Millipore Sigma, Catalog # 32526), ivermectin (12 nM) (Sigma-Aldrich, Catalog # I8898-1G), fenbendazole (10.65 μM) (Sigma-Aldrich, Catalog # F5396-5G), mebendazole (48 μM) (Sigma-Aldrich, Catalog # M2523-25G), milbemycin oxime (120 nM) (Millipore Sigma, Catalog # 1443806), moxidectin (3 nM) (Millipore Sigma, Catalog # 113507-06-5), ricobendazole (25 μM) (Santa Cruz Biotechnology, Catalog # sc-205838), selamectin (0.39 μM) (Sigma-Aldrich, Catalog # SML2663-25MG), and thiabendazole (62.99 μM) (Sigma-Aldrich, Catalog # T5535-50G) using the methods as described in *High-throughput assays (HTAs)*. After measuring nematode responses, phenotypic data were cleaned and processed as described in *High-throughput assay (HTA) data collection and data cleaning*. Wild strains that lacked phenotype measurements for one or more drugs were removed from the dataset prior to statistical analysis.

Spearman rank-order correlation and significance testing were performed using the R package *Hmisc* (version 4.1.1). Subsequently, hierarchical clustering was performed using the R package *pheatmap* (version 1.0.12). Significant correlations ($p < 0.05$) were recorded (**S3 Table**). The resulting heat map and dendrogram (**Fig 5**) were constructed using Euclidean distance and complete linkage metrics, and split into their two largest clusters.

### Neuronal expression patterns of genes encoding GluCls and beta-tubulin

Single-cell RNA-sequencing data were obtained from the Complete Gene Expression Map of the *C. elegans* Nervous System (CeNGEN) [106]. Using the CeNGEN scRNA-seq dataset, gene expression for each of the genes of interest was extracted from the database with a threshold of 2 (**S8 Table**). All expression values are in transcripts per million (TPM) [119]. All data collection, processing, normalization, and analysis of the CeNGEN data can be found at https://www.cengen.org/.

### Gene models for *ben-1* and the three genes encoding for GluCl subunits

Gene models of *ben-1*, *avr-14*, *avr-15*, and *glc-1* were created with a modified script retrieved from the Gene Model Visualization Kit (https://github.com/AndersenLab/GMVK). Gene models physical positions were extracted from WormBase (WS283) [120]. The location of each gene deletion is denoted beneath each gene model.

## Supporting information

**S1 Table. A list of strains and genotypes, along with primer and guide RNA sequences.**
(CSV)

**S2 Table. High-throughput assay (HTA) data.**
(CSV)

**S3 Table. *p*-values from correlation matrices.**
(CSV)

**S4 Table. Anthelmintic drugs, concentrations used, and manufacturer's details.**
(CSV)

**S5 Table. Competitive fitness assay data for DMSO and albendazole.**
(CSV)

**S6 Table. Competitive fitness assay data for DMSO and ivermectin.**
(CSV)

**S7 Table. Fecundity assay data.**
(CSV)

**S8 Table. Cell types expressing beta-tubulin and GluCl subunit genes from CeNGEN.**
(CSV)

**S1 Fig. The competitive fitness assay allows for the assessment of allele frequency in the presence of DMSO, albendazole, or ivermectin. (A)** Equal numbers of the control strain PTM229 were placed on each test plate along with an edited strain. **(B)** Strains were grown on 6 cm NGMA plates for approximately seven days. After seven days, a ~0.5 cm$^3$ plate chunk of NGMA with animals was transferred to a new 6 cm NGMA plate. Animals were washed off of NGMA plates at each odd generation. After animal collection, DNA extractions, DNA cleanup, and quantification were performed. Allele frequencies were quantified using ddPCR. See Methods, *Competitive fitness assays* for details. Modified from a previous version [73]. Created with Biorender.com.
(TIF)

**S2 Fig. Competitive fitness assays across seven generations in DMSO. (A)** A barcoded N2 wild-type strain, PTM229, was competed with strains that have deletions in either one, two, or three genes that encode for GluCl channels or in the beta-tubulin gene *ben-1* in DMSO. Generation is shown on the x-axis, and the relative allele frequencies of the nine strains with genome-edited alleles and N2 are shown on the y-axis. **(B)** The log$_2$-transformed competitive fitness of each allele is plotted in DMSO. The gene tested is shown on the x-axis, and the competitive fitness is shown on the y-axis. Each point represents a biological replicate of that competition experiment. Data are shown as Tukey box plots with the median as a solid horizontal line, and the top and bottom of the box representing the 75th and 25th quartiles, respectively. The top whisker is extended to the maximum point that is within the 1.5 interquartile range from the 75th quartile. The bottom whisker is extended to the minimum point that is within the 1.5 interquartile range from the 25th quartile. Significant differences between the wild-type N2 strain and all the other alleles are shown as asterisks above the data from each strain ($p > 0.05$ = ns, $p < 0.001$ = ***, $p < 0.0001$ = ****, Tukey HSD).
(TIF)

**S3 Fig. Variation in daily fecundity of C. elegans deletion strains in DMSO.** Boxplots for daily fecundity when exposed to DMSO on the y-axis, for each deletion strain on the x-axis. Each point represents the daily fecundity count for one biological replicate. Error bars show the standard deviation of lifetime fecundity among 7–10 biological replicates. Data are shown as Tukey box plots with the median as a solid horizontal line, and the top and bottom of the box represent the 75th and 25th quartiles, respectively. Significant differences between the wild-type strain, N2, and all other deletions are shown as asterisks above the data from each strain ($p > 0.05$ = ns, $p < 0.05$ = *, $p < 0.01$ = **, $p < 0.001$ = ***, $p < 0.0001$ = ****, Tukey HSD).
(TIF)

**S4 Fig. Variation in daily fecundity of *C. elegans* deletion strains in albendazole.** Boxplots for daily fecundity when exposed to albendazole on the y-axis, for each deletion strain on the x-axis. Each point represents the daily fecundity count for one biological replicate. Error bars show the standard deviation of lifetime fecundity among 7–10 biological replicates. Data are shown as Tukey box plots with the median as a solid horizontal line, and the top and bottom of the box represent the 75th and 25th quartiles, respectively. Significant differences between the wild-type strain, N2, and all other deletions are shown as asterisks above the data from

each strain ($p > 0.05$ = ns, $p < 0.05$ = *, $p < 0.01$ = **, $p < 0.001$ = ***, $p < 0.0001$ = ****, Tukey HSD).
(TIF)

**S5 Fig. Variation in daily fecundity of *C. elegans* deletion strains in ivermectin.** Boxplots for daily fecundity when exposed to ivermectin on the y-axis, for each deletion strain on the x-axis. Each point represents the daily fecundity count for one biological replicate. Error bars show the standard deviation of lifetime fecundity among 7–10 biological replicates. Data are shown as Tukey box plots with the median as a solid horizontal line, and the top and bottom of the box represent the 75th and 25th quartiles, respectively. Significant differences between the wild-type strain, N2, and all other deletions are shown as asterisks above the data from each strain ($p > 0.05$ = ns, $p < 0.05$ = *, $p < 0.01$ = **, $p < 0.001$ = ***, $p < 0.0001$ = ****, Tukey HSD).
(TIF)

**S6 Fig. High-throughput assays for each deletion strain in control conditions.** Median animal length values from populations of nematodes grown in DMSO are shown on the y-axis. Each point represents the median animal length from a well containing approximately 5–30 animals. Data are shown as Tukey box plots with the median as a solid horizontal line, the top and bottom of the box representing the 75th and 25th quartiles, respectively. The top whisker is extended to the maximum point that is within 1.5 interquartile range from the 75th quartile. The bottom whisker is extended to the minimum point that is within 1.5 interquartile range from the 25th quartile. Significant differences between the wild-type strain and all other strains are shown as asterisks above the data from each strain ($p > 0.05$ = ns, $p < 0.05$ = *, $p < 0.001$ = ***, $p < 0.0001$ = ****, Tukey HSD).
(TIF)

**S7 Fig. High-throughput assays for each deletion strain in 500 nM of ivermectin.** The regressed median animal length values for populations of nematodes growth in 500 nM ivermectin are shown on the y-axis. Each point represents the regressed median animal length value of a well containing approximately 5–30 animals. Data are shown as Tukey box plots with the median as a solid horizontal line, and the top and bottom of the box representing the 75th and 25th quartiles, respectively. The top whisker is extended to the maximum point that is within the 1.5 interquartile range from the 75th quartile. The bottom whisker is extended to the minimum point that is within the 1.5 interquartile range from the 25th quartile. Significant differences between the wild-type strain and all other deletions are shown as asterisks above the data from each strain ($p > 0.05$ = ns, $p < 0.001$ = ***, $p < 0.0001$ = ****, Tukey HSD).
(TIF)

**S8 Fig. Upset plot of the neuronal expression patterns for *ben-1*, *avr-14*, *avr-15*, and *glc-1*.** Upset plot of single-cell RNA-sequencing data obtained from CeNGEN. Horizontal bar plots sum the total number of neurons where the gene is expressed. Vertical bar plots sum overlap where genes are expressed in the neuronal cell subtypes. Black dots directly under vertical bar plots signify the gene(s) that overlap in the neuronal cell subtypes indicated in the vertical bar plot.
(TIF)

**S9 Fig. Dose response of the avr-15 deletion in ivermectin on NGMA.** The single deletion strain *avr-15* was exposed to ivermectin in a dose response manner. Animals were exposed to ivermectin at **(A)** 0 nM (1% DMSO), **(B)** 1.2 nM, **(C)** 1.3 nM, **(D)** 1.4 nM, **(E)** 1.5 nM, **(F)** 1.6 nM, **(G)** 1.7 nM, **(H)** 1.8 nM, **(I)** 1.9 nM, **(K)** 2.1 nM, **(L)** 2.25 nM, **(M)** 2.5 nM, **(N)** 5 nM, and

**(O)** 10 nM.
(TIF)

**S10 Fig. Dose response of the triple mutant deletion strain in ivermectin on NGMA.** The triple mutant deletion strain (*avr-14*, *avr-15*, and *glc-1*) was exposed to ivermectin in a dose response manner. Animals were exposed to ivermectin at **(A)** 0 nM (1% DMSO), **(B)** 1.2 nM, **(C)** 1.3 nM, **(D)** 1.4 nM, **(E)** 1.5 nM, **(F)** 1.6 nM, **(G)** 1.7 nM, **(H)** 1.8 nM, **(I)** 1.9 nM, **(K)** 2.1 nM, **(L)** 2.25 nM, **(M)** 2.5 nM, **(N)** 5 nM, and **(O)** 10 nM.
(TIF)

## Acknowledgments

We would like to thank members of the Andersen laboratory for their feedback and helpful comments on this manuscript. We thank SunyBiotech for providing us with the single GluCl deletion strains. We thank the *Caenorhabditis* Natural Diversity Resource (NSF Capacity grant 2224885) for providing us with strains used in the wild strain HTA. We thank Joy N. Nyaanga, Timothy A. Crombie, and Samuel J. Widmayer for creating *easyXpress*.

## Author Contributions

**Conceptualization:** Erik C. Andersen.

**Data curation:** Amanda O. Shaver.

**Formal analysis:** Amanda O. Shaver.

**Funding acquisition:** Erik C. Andersen.

**Investigation:** Amanda O. Shaver.

**Methodology:** Amanda O. Shaver, Isabella R. Miller, Etta S. Schaye, Nicolas D. Moya, J. B. Collins, Janneke Wit, Alyssa H. Blanco, Fiona M. Shao, Elliot J. Andersen, Sharik A. Khan, Gracie Paredes.

**Project administration:** Erik C. Andersen.

**Resources:** Erik C. Andersen.

**Software:** Amanda O. Shaver.

**Supervision:** Erik C. Andersen.

**Validation:** Amanda O. Shaver.

**Visualization:** Amanda O. Shaver, Etta S. Schaye, Nicolas D. Moya.

**Writing – original draft:** Amanda O. Shaver.

**Writing – review & editing:** Amanda O. Shaver, Isabella R. Miller, Etta S. Schaye, Nicolas D. Moya, J. B. Collins, Janneke Wit, Erik C. Andersen.

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
