## [Decision Letter · Decision Letter 0]

15 Mar 2024

Dear Dr. Andersen,

Thank you very much for submitting your manuscript "Quantifying the fitness effects of resistance alleles with and without anthelmintic selection pressure using Caenorhabditis elegans" for consideration at PLOS Pathogens. As with all papers reviewed by the journal, your manuscript was reviewed by members of the editorial board and by several independent reviewers. In light of the reviews (below this email), we would like to invite the resubmission of a significantly-revised version that takes into account the reviewers' comments.

Three qualified reviewers have offered markedly different recommendations for disposition of this manuscript. All agree that it is well-written and that it contains important information, although one reviewer questioned the relevance of the data for parasitic nematodes. After carefully reading the manuscript and the reviews, I believe the proper course of action is to request a revision in which each of the comments raised in the review process are thoroughly addressed or refuted. The most significant and common concern is that null mutations in AVR receptors have not been encountered in field-derived populations of parasitic nematodes resistant to macrocyclic lactones, which requires a thoughtful response from the authors. I believe that most of the other concerns can be more readily resolved.

We cannot make any decision about publication until we have seen the revised manuscript and your response to the reviewers' comments. Your revised manuscript is also likely to be sent to reviewers for further evaluation.

Sincerely,

Timothy G. Geary, PhD

Guest Editor

PLOS Pathogens

James Collins III

Section Editor

PLOS Pathogens

Michael Malim

Editor-in-Chief

PLOS Pathogens

orcid.org/0000-0002-7699-2064

Three qualified reviewers have offered markedly different recommendations for disposition of this manuscript. All agree that it is well-written and that it contains important information, although one reviewer questioned the relevance of the data for parasitic nematodes. After carefully reading the manuscript and the reviews, I believe the proper course of action is to request a revision in which each of the comments raised in the review process are thoroughly addressed or refuted. The most significant and common concern is that null mutations in AVR receptors have not been encountered in field-derived populations of parasitic nematodes resistant to macrocyclic lactones, which requires a thoughtful response from the authors. I believe that most of the other concerns can be more readily resolved.

Reviewer's Responses to Questions

**Part I - Summary**

Reviewer #1: The manuscript by Shaver and colleagues describes the use of a C. elegans model to explore the phenotypes conferred by loss of function mutations in known anthelmintic target genes, with and without drug exposure.

I found the approach and resulting data very interesting – the competitive fitness assays and comparisons between wild isolates were especially well done. The finding of significant fitness costs associated with loss of function mutations in the GluCl subunits is a nice observation that has long been speculated to be the case, and fits with recent findings that these targets are not implicated in resistance in parasitic nematodes. However, I felt the introduction and discussion could be improved as the relevance and suggested implications for resistance in parasitic nematodes were not always well reasoned. I have only minor edits for the results section and methods, but I suggest the introduction and discussion should be re-drafted for clarity and accuracy.

Reviewer #2: The manuscript aims at understanding the mechanisms and consequences of resistance to albendazole and ivermectin drugs, especially when administered together. While Albendazole resistance is mainly linked to variations in the beta-tubulin gene, ivermectin targets GluCl genes, but it is unknown whether GluCl genes are involved in ivermectin resistance in nature. Therefore, the study, conducted with Caenorhabditis elegans, investigated the fitness costs associated with the loss of drug target genes. The results revealed that albendazole resistance required the deletion of the beta-tubulin gene ben-1, while ivermectin resistance necessitated the loss of two (avr-14 and avr-15) or three (avr-14, avr-15, and glc-1) GluCl genes. Fecundity assays showed no fitness benefit in albendazole with the loss of ben-1, and no GluCl mutants were resistant to ivermectin. The study also explored evidence of multi-drug resistance across different traits, finding no evidence that loss of ben-1 or any GluCl subunit conferred cross-resistance.

The manuscript is well written and the materials and methods is very detailed. Moreover, the high-throughput assay (HTA), rigorously conducted, is an original way of assessing nematode fitness. While the conclusions about benzimidazole resistance are very clear, the relevance of the results on ivermectin resistance is less so. Indeed, while the authors findings provide insights into why GluCls have not been clearly associated with resistance in parasites, it does not answer how resistance alleles could spread in parasitic nematodes populations since mutations of GluCl are not found in selected ivermectin resistant strains, although the authors have set themselves the goal of answering this question. Instead, the authors could have used established resistant strains like the IVR10 (James et al. International Journal for Parasitology, 2009) or strains deleted for genes required for amphids formation that are naturally resistant to MLs (such as the dyf-7 deleted strain).

Reviewer #3: The manuscript provides novel and detailed insight into the fitness effects upon mutation of genes associated with anthelmintic resistance in the model nematode C. elegans. Defining fitness costs helps predict potential mechanisms of resistance against ivermectin and albendazole in natural settings. The authors have measured multiple aspects of fitness in mutants strains, mostly in large experiments of which the results deepen our understanding of previously identified genes involved in anthelmintic resistance. Finally, the authors present a detailed discussion of the (potential) consequences of their results for parasitic nematode species. The manuscript is well-written and adds substantial experimental evidence for understanding mechanisms of anthelmintic resistance.

**Part II – Major Issues: Key Experiments Required for Acceptance**

Reviewer #1: (No Response)

Reviewer #2: Figure 4: The authors state: “we did not see a significant difference in median animal length between the double GluCl mutant avr-14 and avr-15 or the triple GluCl mutant, as reported previously [60]”. This is quite surprising because according to Dent et al. the triple mutant is supposed to be at least 40-fold more resistant than the double avr-14 and avr-15 mutant. Moreover, Glendinning et al. PLOS One, 2011, showed that the sensitivity to Ivermectin of the triple mutatant strain DA1316 can be restored by re-expressing avr-14 or other GluCl subunits, which confirm Dent et al. results. One main difference between the present work and the experiments conducted by Dent at al. is that here the authors performed their assays in liquid media and only with two doses of ivermectin. To draw definitive answers about the possible differential ivermectin sensitivity between the double and the triple mutants, could the authors perform Larval development assays with multiple ivermectin doses (dose/response) both on NGM plates and liquid media? This should confirm either their original results or the observations made by Dent et al.

Reviewer #3: - Mutations (avr-14, avr-15 and glc-1) in the GluCl units of C. elegans were created in the strain PD1074, but were experimentally compared to the N2 strain and ben-1 mutant in the N2 background. Although the PD1074 is derived from the N2 strain, it is not completely clear how they (may) differ in fitness and whether this could affect the experiments by the authors. The authors should clarify based on either previous results / literature (like they explicitly do for strain PTM229) or their own experimental data if and how differences between N2 and PD1074 may affect interpretation of results.

- Line 706 (HTA method) describes that wells with fewer than 5 animals were excluded from the analysis. This is logical in terms of acquiring median and mean lengths which is not possible when too few or no animals are present. However, if embryos of highly sensitive strains do not hatch, biologically relevant information may be excluded from the assay. Principally this data could also be extracted by readers from Table S2. Although it’s highly appreciated the raw data is included with the manuscript, it would help to summarise information about (potential) overrepresentations of excluded strains.

**Part III – Minor Issues: Editorial and Data Presentation Modifications**

Reviewer #1: Line 89 ‘However, anthelmintic resistance has emerged’ - clarify this is in veterinary helminths with only reduced efficacy reported in human helminths.

96 – ‘when a single anthelmintic has reduced efficacy’ – misleading as combination treatment needs to be initiated while both anthelmintics are highly effective (in veterinary helminths, combination treatments have not been effective after resistance has emerged)

99 – ‘despite the successes of both strategies’ – combination treatment and rotation are both widely used in livestock and horses but have failed to control resistance

104 – ‘however, neither co-administration nor rotation alone are enough to slow the spread of resistance’ – I don’t believe we have the data to support this statement, we only know this approach fails when resistance to one or more of the anthelmintics is already present – co-administration may be effective approach to preserve new anthelmintics

168 – add ‘at these particular loci’

222 – ‘incredibly’ to ‘severely’

Figs 2 and 3 – helpful to add drug concentrations to legend

285 – 325 fecundity data was very variable – is it possible the drug concentrations chosen were slightly too low to show a clear phenotype (e.g. no decrease in fecundity in N2 reference for either drug) and is there any possibility the regular transfers of hermaphrodites to new plates could reduce their viability/fecundity in a variable manner depending on transfer technique? The many conclusions from these experiments were quite unexpected and sometimes contradictory (e.g. avr-15 and avr15:avr14 LOF mutants were less fecund overall but both were significantly more fecund than controls and all other lines from days 5-7 in all conditions) which makes it difficult to assess if the interpretations are robust.

338 - 340 unclear what this statement means

376 – 380 points are valid but could this be phrased better?

418 – 419 ‘to identify why GluCls in parasites lack variation’ what does this mean and please add references

430 ‘than what was tested here’ rephrase

447 – 448 could add ‘but more recent genome-wide analyses have confirmed that mutations in GluCl subunit genes are not implicated in ivermectin resistance in Haemonchus’

453 ‘lack of variation’ again needs clarified and references

467 paper cited does not show ivermectin resistance involves ‘various genes and mechanisms’

489 – 490 needs rephrased

496 ‘the overlap in expression in the same neurotransmitters’ does not make sense

514 – 515 ‘our understanding of ivermectin’s MoR remains incomplete because we have not identified all the genes involved in ivermectin resistance’ need to add ‘in C. elegans, or any genes involved in ivermectin resistance in parasitic nematodes’

Reviewer #2: 1) In the introduction, the paragraph on the co-administration of two or three drugs (lanes 95-101) seems to me to be oversold, and the bibliographical references mentioned do not fully support the authors' assertions.

2) Lane 112: The claim that Bz and ML are “regularly Co-administrated” seems to be an overstatement. The authors give an old reference that do not support this claim”. Lane 368: The authors claim that: “Because albendazole and ivermectin are routinely distributed together to at-risk populations, it is critical to ensure that the two drugs do not have the same MoR to avoid the possibility of cross-resistance”. To my knowledge, this practice is not very widespread. Can the authors provide appropriate bibliographical references to back up their assertions?

3) Figure 2E: Could the authors give an explanation to the sharp decrease in allele frequency after one day for both the avr-14 and avr-15 and the triple mutant strain in the competitive fitness assay carried in presence of ivermectin? Also, this is very surprising to observe fixation only two days later for these strains.

4) Figure 3: I agree with the authors that: “Because strains with a loss of both avr-14 and avr-15 and a loss of all three GluCl subunits have significantly reduced fecundity across all conditions, it would be unlikely for animals in nature to acquire loss-of-function mutations in these genes that cause detrimental fitness consequences”. However, could the authors explain why fecundity (Figure 2) and comparative fitness (Figure 3) are sometimes correlated and other times not? Are strain life-

---

## [Editor Report · Decision Letter 1]

7 May 2024

Dear Dr. Andersen,

We are pleased to inform you that your manuscript 'Quantifying the fitness effects of resistance alleles with and without anthelmintic selection pressure using Caenorhabditis elegans' has been provisionally accepted for publication in PLOS Pathogens.

Best regards,

Timothy G. Geary, PhD

Guest Editor

PLOS Pathogens

James Collins III

Section Editor

PLOS Pathogens

Michael Malim

Editor-in-Chief

PLOS Pathogens

orcid.org/0000-0002-7699-2064

I deeply appreciate the constructive, thoughtful and positive responses of the authors to the concerns raised by the reviewers. The changes made as a result in the revised version of the manuscript and the explanations provided by the authors have completely and convincingly resolved these concerns, and the manuscript, in my judgment, should noe be processed for publication.
---

## [Editor Report · Acceptance letter]

13 May 2024

Dear Dr. Andersen,

We are delighted to inform you that your manuscript, "Quantifying the fitness effects of resistance alleles with and without anthelmintic selection pressure using Caenorhabditis elegans," has been formally accepted for publication in PLOS Pathogens.

Best regards,

Michael Malim

Editor-in-Chief

PLOS Pathogens

orcid.org/0000-0002-7699-2064